# Cytomegalovirus vaccine vector-induced effector memory CD4 + T cells protect cynomolgus macaques from lethal aerosolized heterologous avian influenza challenge

An influenza vaccine approach that overcomes the problem of viral sequence diversity and provides long-lived heterosubtypic protection is urgently needed to protect against pandemic influenza viruses. Here, to determine if lung-resident effector memory T cells induced by cytomegalovirus (CMV)-vectored vaccines expressing conserved internal influenza antigens could protect against lethal influenza challenge, we immunize Mauritian cynomolgus macaques (MCM) with cynomolgus CMV (CyCMV) vaccines expressing H1N1 1918 influenza M1, NP, and PB1 antigens (CyCMV/Flu), and challenge with heterologous, aerosolized avian H5N1 influenza. All six unvaccinated MCM died by seven days post infection with acute respiratory distress, while 54.5% (6/11) CyCMV/Flu-vaccinated MCM survived. Survival correlates with the magnitude of lung-resident influenza-specific CD4 + T cells prior to challenge. These data demonstrate that CD4 + T cells targeting conserved internal influenza proteins can protect against highly pathogenic heterologous influenza challenge and support further exploration of effector memory T cell-based vaccines for universal influenza vaccine development.

The world remains at risk of another influenza pandemic. The four influenza pandemics of the past 100 years killed tens of millions of people, yet a universal influenza vaccine capable of protecting against future pandemic influenza viruses still does not exist. Current antibody-mediated influenza vaccines are strain-specific due to targeting of the highly variable hemagglutinin (HA) and neuraminidase (NA) glycoproteins. Indeed, given the continual sequence evolution of HA and NA through antigenic drift and ability of the segmented virus to recombine two or more different strains through antigenic shift, seasonal influenza vaccine effectiveness ranges from 30% to 60% depending on matching of the vaccine sequence to influenza viruses subsequently circulating that year[1,2].

Furthermore, these vaccines provide little, if any, protection against pandemic influenza viruses. Of particular concern are highly pathogenic avian influenza (HPAI) viruses circulating among wild and captive birds, such as H5N1, which has a documented fatality rate of 52% in humans[3]. Given the increasing number and geographical spread of HPAI infections in birds[4], and the minimal amino acid changes necessary for avian influenza viruses to become transmissible via aerosol droplets in mammals[5–7], the world is at severe risk of an HPAI pandemic. Indeed, the first case of mammal to human transmission of H5N1 was recently reported in a dairy farm worker[8], highlighting the potential for HPAI to transmit to humans. Thus, new vaccine approaches capable of protecting against all

✉e-mail: smbb@pitt.edu; dsreed@pitt.edu; sacha@ohsu.edu

influenza strains, and particularly against HPAI with pandemic potential, are urgently needed.

While current antibody-based vaccine approaches can provide sterilizing immunity, they narrowly focus on subtype-specific HA and NA sequences and are thus susceptible to antigenic mismatch with new strains that arise annually via antigenic drift or emerge suddenly with pandemic potential via genetic reassortment. In contrast to the high sequence diversity of influenza HA and NA glycoproteins, the internal structural proteins such as matrix (M) and nucleocapsid protein (NP), and viral polymerases like PB1, are highly conserved across all strains[9]. Indeed, pre-existing T cells targeting internal influenza proteins can recognize disparate influenza variants and provide heterosubtypic protection against disease against novel influenza strains in humans[10–12]. Harnessing T cell immunity against internal influenza proteins therefore represents a potential pathway towards universal influenza vaccine development. However, there is a dearth of vaccine vectors capable of priming and maintaining the high frequencies of pulmonary influenza-specific effector memory T cells ($T_{EM}$) likely needed for protection. Indeed, almost all currently utilized clinical vaccine platforms, including the current whole inactivated influenza virus and messenger RNA delivery approaches, induce pathogen-specific T cells with a predominantly central memory ($T_{CM}$) phenotype that require a period of anamnestic expansion prior to exerting antiviral activity following infection[13,14]. In contrast, the β-herpesvirus cytomegalovirus (CMV) elicits high-frequency $T_{EM}$ that home to peripheral organs, particularly the lung, where they are pre-positioned to intercept pathogens soon after infection[15], suggesting that CMV may be an ideal vector for development of an influenza-specific T cell-based vaccine.

Despite differences in their etiology, both influenza and HIV possess viral glycoproteins with high sequence diversity that stymie broadly-neutralizing antibody-based vaccine development, and, therefore, vaccine advancements that circumvent glycoprotein diversity in one virus may be transferrable to the other[16]. A pre-clinical HIV vaccine approach based on strain 68-1 rhesus CMV expressing internal simian immunodeficiency virus (SIV) antigens (RhCMV/SIV) elicits high-frequency SIV-specific $T_{EM}$ that control SIV replication in 59% of vaccinated rhesus macaques (RM) across multiple studies in the absence of vaccine-elicited antibodies[17]. Strain 68-1 RhCMV, which lacks the pentameric receptor complex components Rh157.5/Rh157.4 and the viral CXC chemokine-like Rh158-Rh161 gene products (orthologs of human CMV [HCMV] UL128/UL130 and UL146/UL147 genes, respectively), induces MHC-E- and MHC-II-restricted CD8 + T cells[17,18]. However, while strain 68-1 RhCMV elicits unconventionally MHC-E- or MHC-II- restricted CD8 + T cells, RhCMV vectors can be genetically modified via repair of the Rh157.5/Rh157.4 and Rh158-Rh161 genes to generate full length (FL) RhCMV vectors that elicit conventionally MHC-Ia-restricted CD8 + T cells[19]. Therefore, CMV vectors can be generated to elicit the type of MHC-restricted CD8 + T cell required for protection against a particular pathogen. Indeed, while SIV-specific MHC-E-restricted CD8 + T cells are required for RhCMV/SIV-mediated protection against SIV replication[17,20,21], they are not required for RhCMV/Mycobacterium tuberculosis (MTB)-mediated protection against MTB in RM[22]. Regardless of MHC-restriction, RhCMV-induced CD4+ and CD8 + T cells persist longitudinally for years after the initial vaccination and accumulate to high levels in lung[23–25]. Finally, given the unique protection against SIV replication observed in RhCMV/SIV-vaccinated RM, clinical trials are currently underway to test the safety and immunogenicity of a HCMV vaccine vector for HIV[26], providing a potential pathway for clinical CMV-based vaccines against other pathogens. Based on these attributes, we hypothesized that a CMV-based vaccine expressing conserved internal influenza proteins would generate high-frequency, pulmonary-resident, influenza-specific $T_{EM}$ with the ability to protect against a HPAI isolate with pandemic potential such as H5N1.

# Results

## Generation and immunogenicity of CyCMV vaccine vectors for influenza

To determine whether a CMV-based vaccine could protect against HPAI, we elected to utilize a stringent model of aerosolized H5N1 challenge of Mauritian cynomolgus macaques (MCM) where infection is uniformly lethal[27]. Given the strict species-specificity of CMV, strain 68-1 RhCMV does not infect MCM and cannot elicit T cell responses[28], thereby precluding its use as a vaccine vector in the MCM model of aerosolized influenza. To facilitate use of MCM for CMV-based experiments, we recently isolated and characterized a full length cynomolgus macaque CMV (FL CyCMV) isolate, and subsequently generated a strain "68-1 like" double deleted (dd CyCMV) vaccine vector, whereby the CyCMV orthologues of the RhCMV pentameric receptor complex components Rh157.5/Rh157.4 and the viral CXC chemokine-like Rh158-Rh161 gene products are deleted to reflect genetic deletions present in strain 68-1 RhCMV[29]. Vaccination of MCM with FL CyCMV expressing SIV Gag elicited Gag-specific MHC-Ia-restricted CD8 + T cells, while vaccination of MCM with dd CyCMV expressing SIV Gag generated Gag-specific CD8 + T cells that were either MHC-II- or MHC-E-restricted, mirroring the MHC restriction patterns elicited by FL or strain 68-1 RhCMV/Gag in RM, respectively[29]. Furthermore, half of dd CyCMV/ SIV-vaccinated MCM controlled SIV replication post infection, and manifested a vaccine-induced IL-15 transcriptomic signature that is associated with efficacy in RhCMV/SIV-vaccinated RM[29,30]. Therefore, central features of the RhCMV vaccine vector in RM are conserved with CyCMV in MCM, facilitating pre-clinical studies in MCM-based models of disease.

In addition to MHC-Ia, the major cellular targets of influenza virus, epithelial cells and type I and II pneumocytes, express MHC-E and MHC-II[31–34]. Therefore, we generated two sets of CyCMV vaccine vectors expressing influenza antigens, one set based on FL CyCMV to induce MHC-Ia-restricted CD8 + T cells, and another based on dd CyCMV to induce MHC-II- and MHC-E-restricted CD8 + T cells (Supplementary Fig. 1A–C). We selected 1918 H1N1 influenza M, NP, and PB1 as vaccine antigens for the following two reasons: 1) in contrast to the more variable HA and NA glycoproteins, the M, NP, and PB1 proteins are highly conserved among the human and avian influenza viruses recorded across the previous decades[35], making them ideal T cell targets (Supplementary Fig. 1D, and 2) using 1918 influenza antigen sequences would yield nearly a century of natural global influenza evolution between the 1918 influenza vaccine antigen sequences and the heterologous influenza A/Vietnam/1203/2004 (H5N1) challenge virus, facilitating a stringent test of the protective capabilities of a T cell-based vaccine for influenza. We generated three separate vectors by inserting the 1918 influenza M1, NP, or PB1 sequence into the Cy110 open reading frame (ORF) of either FL or dd CyCMV, and confirmed protein expression in each vector in infected fibroblasts in vitro, to generate a set of vaccine vectors collectively named FL CyCMV/Flu or dd CyCMV/Flu, respectively (Supplementary Fig. 1A–C). The Cy110 gene, the orthologue of the HCMV UL82 gene encoding the pp71 protein necessary for lytic replication, was selected for the site of antigenic insertion as this configuration retains genome and transgene stability while rendering a spread-deficient CMV vector with an increased safety profile that maintains immunogenicity[30].

We vaccinated six influenza seronegative MCM subcutaneously with $1 \times 10^7$ PFU with each of three individual vectors comprising either FL CyCMV/Flu or dd CyCMV/Flu and boosted with the same dose at 15 weeks post prime Fig. 1A; Supplementary Table 1). One FL CyCMV/ Flu vaccinated MCM died from study-unrelated causes at 129 days post immunization, leaving five MCM in that group for all timepoints onward. We monitored the influenza transgene-specific CD4+ and CD8 + T cell response in peripheral blood and found that these responses persisted throughout the vaccine induction phase with no

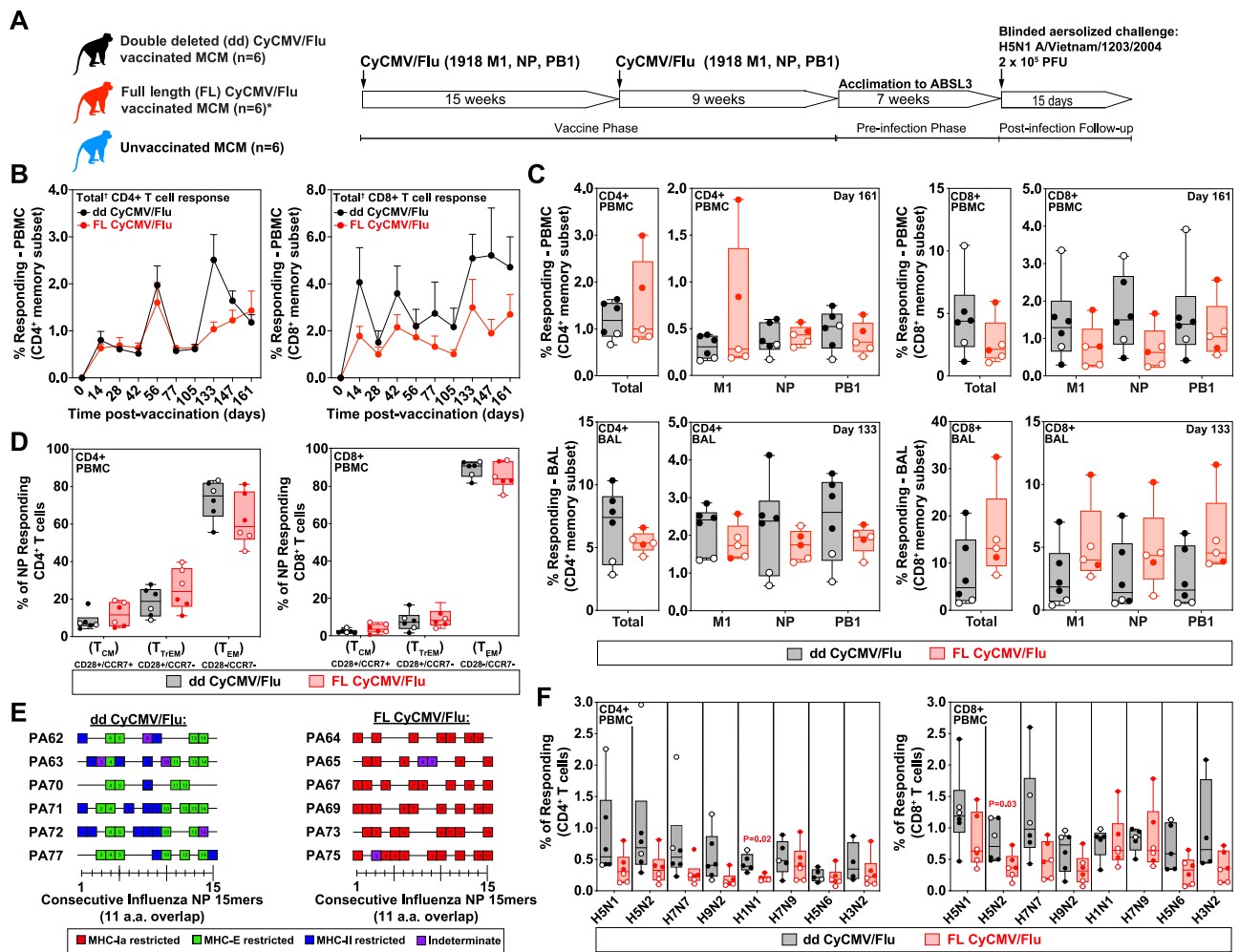

**Fig. 1 | Vaccine phase immunology of CyCMV/Flu-vaccinated Mauritian cyno-molgus macaques (MCM). A** Study overview with major events illustrated. *Note that one FL CyCMV/Flu-vaccinated MCM died during the vaccine phase due to study-unrelated causes. **B** Total (summed responses against 1918 influenza M1 + NP + PB1) vaccine-induced CD4+ (left) and CD8+ (right) T cell responses during the vaccine phase in dd CyCMV/Flu-vaccinated (black) and FL CyCMV/Flu-vaccinated (red) MCM. N = 6 for each timepoint, except for n = 5 for FL CyCMV/Flu from day 133 onwards due to loss of one animal. **C** Magnitude of the CD4+ (left two graphs) and CD8+ (right two graphs) T cell responses in peripheral blood (top row) and bronchoalveolar lavages (BAL, bottom row) against all antigens summed or individual antigens at the final timepoint prior to H5N1 challenge in dd CyCMV/Flu-vaccinated (black) and FL CyCMV/Flu-vaccinated (red) MCM. Open circles denote MCM that succumbed to challenge while closed circles denote those that survived. N = 6 for dd CyCMV/Flu and n = 5 for FL CyCMV/Flu. **D** Memory phenotype of NP transgene-specific CD4+ (left) and CD8+ (right) T cells in the PBMC of the ddCyCMV/Flu-vaccinated (black) and FL CyCMV/Flu-vaccinated (red) MCM. Open circles denote MCM that succumbed to challenge while closed circles denote those

that survived. PBMC for this assay were pooled from days 28, 56, 161, and 168 post vaccination. N = 6 for each group. **E** MHC restriction of CD8 + T cell responses to 1918 influenza NP. Boxes depict NP 15mers recognized by CD8 + T cells in a given animal with colors indicating MHC-restriction of each response as indicated. PBMC for this assay were pooled from days 105, 119, 133, 147, 161, and 168 post vaccination. **F** Recognition of the various inactivated whole influenza isolates indicated on the x axis by CD4+ (left) or CD8+ (right) T cells in the PBMC of dd CyCMV/Flu-vaccinated (black) and FL CyCMV/Flu-vaccinated (red) MCM following overnight co-culture. Open circles denote MCM that succumbed to challenge while closed circles denote those that survived. N = 6 for each group. PBMC for this assay were pooled from days 105, 119, 133, 147, 161, and 168 post vaccination. Box plots in **B–D, F** show jittered points and a box from first to third quartiles and a line at the median, with whiskers extending to the farthest data point within 1.5x IQR above and below the box. Source data are provided as a Source Data file. Figure 1A created with BioRender.com released under a Creative Commons Attribution-Noncommercial-NoDerivs 4.0 International license.

significant differences between the magnitude of the response engendered by FL or dd CyCMV (Fig. 1B). All three influenza transgenes were recognized by both CD4+ and CD8 + T cell responses, with no significant differences observed between those elicited by FL versus dd CyCMV in peripheral blood or lung, as measured by bronchoalveolar lavages (BAL) (Fig. 1C). In line with previous observations of CMV vaccine vectors in nonhuman primates[22,24,29], high frequencies of transgene-specific CD4+ and CD8 + T cells were observed in the BAL at the final timepoint measured prior to influenza challenge in both FL and dd CyCMV/Flu-vaccinated MCM (Fig. 1C, bottom row). As expected with CMV-based vectors, CyCMV/Flu-elicited CD4+ and CD8 + T cells recognizing influenza antigens exhibited a

predominantly T_{EM} phenotype in peripheral blood as measured by expression of CD28 and CCR7 (Fig. 1D). Next, we defined the MHC-restriction of influenza-specific CD8 + T cells engendered by FL CyCMV or dd CyCMV vectors. To this end, we first performed an intracellular cytokine staining (ICS) assay with the first fifteen 15mer-peptides overlapping by 11 amino acids that span the 1918 NP open reading frame to identify CD8 + T cell responses (Fig. 1E). To define the MHC restriction of peptides eliciting a CD8 + T cell response, we repeated the above ICS assay in the presence of each of the following reagents: the pan MHC-I-blocking antibody W6/32, the leader sequence-derived MHC-E-blocking VL9 peptide, the MHC-II-blocking G46.6 antibody, or isotype control reagents. NP-specific CD8 + T cells in dd CyCMV/Flu-

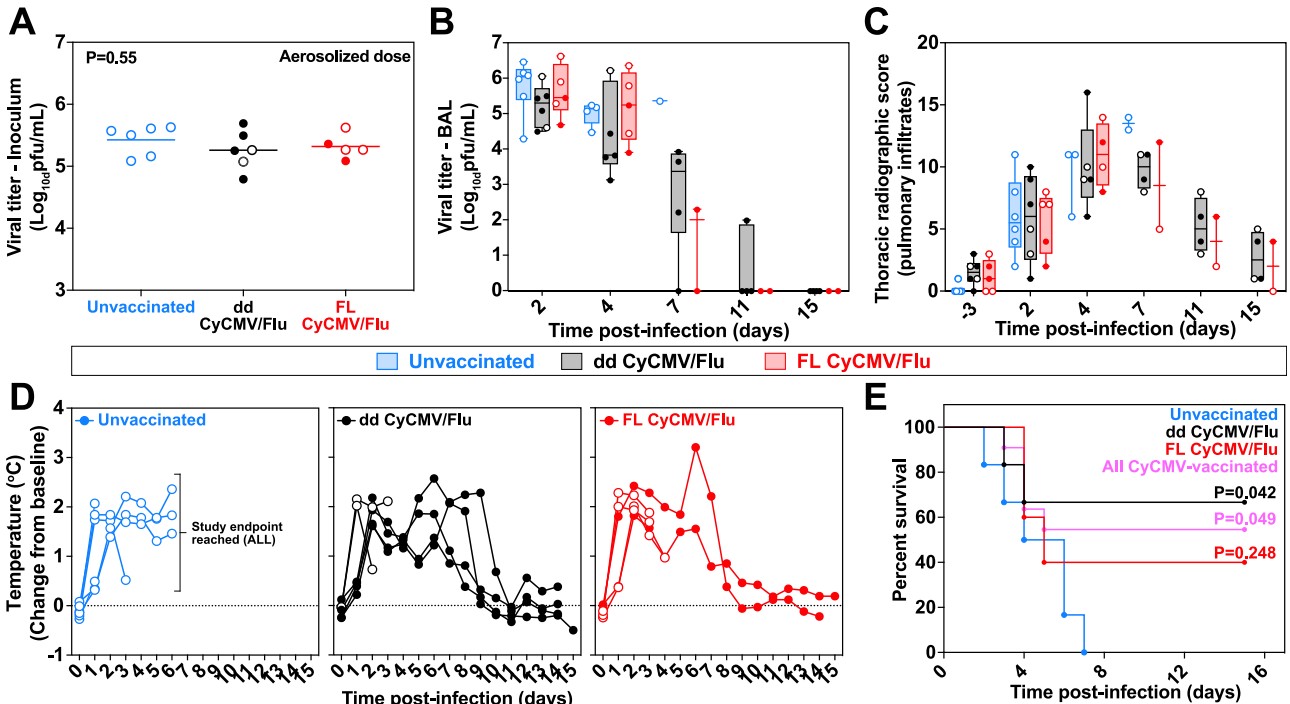

**Fig. 2 | Aerosolized H5N1 challenge of CyCMV/Flu-vaccinated MCM. A** Viral titer of the aerosolized inoculum delivered to each MCM in the unvaccinated (blue), dd CyCMV/Flu (black), or FL CyCMV/Flu (red) group. Open circles denote MCM that succumbed to challenge. $N = 6$ for unvaccinated and dd CyCMV/FL groups, while $n = 5$ for FL CyCMV/Flu due to loss of one animal in vaccine phase. One-way ANOVA overall $p$-value is shown. **B** Viral titer of BAL samples taken after challenge. Open circles denote MCM that succumbed to the infection. $N = 6$ for unvaccinated and dd CyCMV/FL groups, while $n = 5$ for FL CyCMV/Flu at first timepoint. **C** Summed thoracic radiograph score of MCM following challenge. Open circles denote MCM that succumbed to the infection. $N = 6$ for unvaccinated and dd CyCMV/FL groups, while $n = 5$ for FL CyCMV/Flu at first timepoint. **D** Temperature change from baseline of MCM following challenge. Open circles denote MCM that succumbed to the infection. $N = 6$ for unvaccinated and dd CyCMV/FL groups, while $n = 5$ for FL CyCMV/Flu at first timepoint. **E** Survival curve of MCM in the unvaccinated (blue), dd CyCMV/Flu (black), or FL CyCMV/Flu (red) groups. $P$-values shown are log-rank test of each group versus unvaccinated. Box plots in **A**–**C** show jittered points and a box from first to third quartiles and a line at the median, with whiskers extending to the farthest data point within 1.5x IQR above and below the box. Source data are provided as a Source Data file.

vaccinated MCM recognized peptides in the context of MHC-E or MHC-II, while NP-specific CD8 + T cells in FL CyCMV/Flu-vaccinated MCM recognized peptides in the context of MHC-Ia (Fig. 1E). Finally, we measured whether CyCMV/Flu-elicited influenza-specific T cells could recognize diverse influenza isolates in vitro via incubation of PBMC from CyCMV/Flu-vaccinated MCM with inactivated influenza isolates. Both CD4+ and CD8 + T cells from FL and dd CyCMV/Flu-vaccinated MCM responded to all isolates tested, with the majority of isolates recognized similarly by T cells, regardless of the vaccine vector set utilized (Fig. 1F). Thus, both FL and dd CyCMV/Flu-vaccinated MCM generated robust influenza-specific $T_{EM}$ capable of recognizing disparate influenza isolates in vitro, regardless of MHC-restriction, suggesting that these T cell responses might offer protection against challenge with heterologous HPAI.

### Challenge with aerosolized avian H5N1 influenza

To measure the protective capabilities of CyCMV/Flu-induced influenza-specific $T_{EM}$ cells against a heterologous challenge virus, we challenged all MCM with small-particle aerosols containing a target dose of 5.5 log$_{10}$ PFU of the HPAI isolate A/Vietnam/1203/2004(H5N1). To ensure experimental rigor and avoid introduction of any potential bias in clinical scoring, all study staff involved in the aerosolized influenza challenges were blinded to the vaccine status of the MCM until after completion of the challenge phase of the study (Fig. 1A). The inhaled dose was calculated by collecting an aerosol sample during each exposure to measure viral concentration in the aerosol and then multiplying aerosol concentration by the volume of total air inhaled by the MCM during the exposure, as previously described in ref. 27. No

statistical difference in the mean inhaled dose of aerosol virus was measured between the groups (Fig. 2A).

Following challenge with aerosolized influenza, all MCM were monitored for severity of infection via influenza titers in lung via BAL, changes in body temperature via telemetry, pulmonary infiltration via chest radiographs, and development of acute respiratory distress syndrome (ARDS) requiring euthanasia via a pre-defined clinical scoring sheet. As expected, based on experience with CMV-based vectors for SIV and MTB[22,24,29], CyCMV/Flu vaccination did not prevent infection. Infectious influenza virus was found in BAL fluid in every MCM following challenge (Fig. 2B). There was a trend towards lower viral titers in BAL fluid from dd CyCMV/Flu-vaccinated MCM at two and four days post infection, but this difference was not statistically significant. Infectious influenza began to clear from BAL fluid from survivors at day seven post infection, with no virus found in samples from any MCM at 15 days post infection. Longitudinal chest radiographs were performed and scored by radiologists blinded to the treatment of the animals, which revealed pulmonary infiltrates in the lungs of all MCM, with no statistically significant differences observed between the vaccine and control groups (Fig. 2C). All MCM developed fever following infection, which eventually resolved in CyCMV/Flu-vaccinated survivors but not in the unvaccinated controls (Fig. 2D). In line with previous results from this aerosol challenge model[27], all unvaccinated MCM developed ARDS and met humane endpoint criteria within seven days of exposure (Fig. 2E). In contrast, four of six dd CyCMV/Flu-vaccinated MCM survived through the 14-day post-challenge monitoring period, resulting in statistically significant protection from HPAI-induced death. Two out of five FL CyCMV/Flu-vaccinated MCM survived through day 14

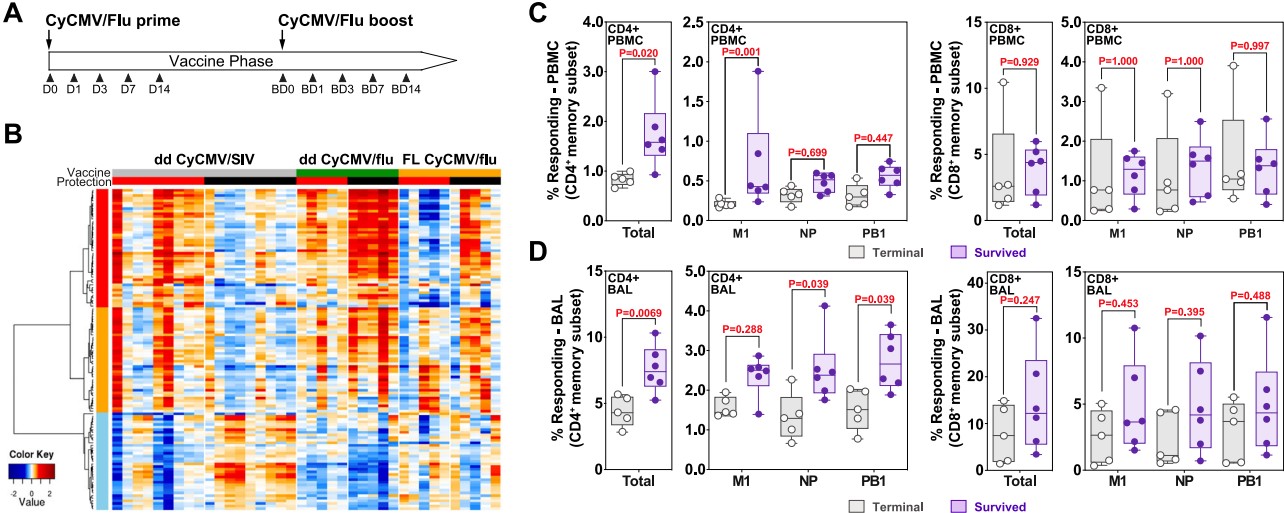

**Fig. 3 | Correlates of protection in CyCMV/Flu-vaccinated MCM protected from lethal H5N1 influenza challenge. A** Experimental timeline of sampling for whole blood transcriptomic analysis during the ddCyCMV/Flu vaccine phase. BD=boost day. **B** Time series heatmap of the dd CyCMV/SIV IL−15 protection signature genes (left, under dd CyCMV/SIV) compared to dd CyCMV/Flu (middle) and FL CyCMV/Flu-vaccinated MCM (right). Protected animals are denoted with a red heading bar versus unprotected animals in black in the protection row. Plotted are log2 fold change values of the 122 leading edge genes across the vaccination time series (Days 0, 1, 3, and 7 post prime and boost immunizations). Red and blue denote up- and down-regulated log fold change to D0, respectively. The dd CyCMV/SIV data are from Malouli et al. [28]. **C** Magnitude of the CD4+ (left two graphs) and CD8+ (right two graphs) T cell responses in peripheral blood against all CyCMV-vectored influenza antigens summed (total, left) or individual antigens (right) at the final timepoint prior to H5N1 challenge in MCM that died (grey open circles) versus

those that survived (lavender closed circles) after H5N1 challenge. $N = 6$ for animals that survived and $n = 5$ for animals that died. PBMC for this assay were from day 161 post vaccination. **D** Magnitude of the CD4+ (left two graphs) and CD8+ (right two graphs) T cell responses in the lung (BAL) against all CyCMV-vectored influenza antigens summed (total, left) or individual antigens (right) at the final timepoint prior to H5N1 challenge in MCM that died (grey open circles) versus those that survived (lavender closed circles) after H5N1 challenge. $N = 6$ for animals that survived and $n = 5$ for animals that died. BAL cells for this assay were from day 133 post vaccination. Box plots in **C** and **D** show jittered points and a box from first to third quartiles and a line at the median, with whiskers extending to the farthest data point within 1.5x IQR above and below the box. In **C** and **D** for total T cell responses a 2-sided $T$ test was used, while for M1, NP, and PB1 comparisons pairwise repeated measures ANOVA with 2-sided Tukey adjustment was used. Source data are provided as a Source Data file.

post exposure, but protection in this group did not reach statistical significance. When both vaccine groups were combined, six of 11 CyCMV/Flu-vaccinated MCM survived, yielding overall vaccine-mediated statistically significant protection against lethal disease, regardless of the specific CyCMV/Flu vaccine vector utilized (Fig. 2E). Thus, while CyCMV-based vaccination did not prevent infection or significantly alter influenza-induced fever and pulmonary infiltration, it did significantly protect against HPAI-induced death.

**Correlates of protection in CyCMV/Flu-vaccinated MCM**
We recently demonstrated that a vaccine-phase, IL-15-based, whole blood transcriptomic signature of protection against SIV replication in strain 68-1 RhCMV/SIV-vaccinated RM was also present in dd CyCMV/SIV-vaccinated MCM that subsequently resisted SIV replication[29,36]. To determine if a similar transcriptomic signature might exist in dd or FL CyCMV/Flu-vaccinated MCM that survived HPAI infection, we performed RNA sequencing of whole blood from all dd and FL CyCMV/Flu-vaccinated MCM immediately prior to vaccination and 1, 3, 7, and 14 days post prime and boost vaccination (Fig. 3A). We then assessed if the previously described RhCMV/SIV protection signature IL-15 response pathway genes enriched in dd CyCMV/SIV-vaccinated MCM could distinguish outcome following HPAI infection. The 122 genes previously identified in CyCMVSIV-mediated protection against SIV replication included genes involved in death receptor signaling, immune cell signaling programs, pattern recognition receptor signaling, and NK cell response (Supplementary Data 1). We plotted the fold change in these genes across the vaccination time series and assessed for conservation of this IL-15-based signature in CyCMV/Flu-vaccinated MCM that survived challenge with otherwise lethal aerosolized HPAI as previously described[29,36]. This analysis revealed that the CyCMV/SIV-

associated protection signature, although present in some vaccinated MCM, did not correlate with CyCMV/Flu-mediated protection against HPAI in either FL or dd CyCMV/Flu-vaccinated MCM, indicating a mechanistically distinct means of protection mediated by CyCMV/Flu against lethal HPAI compared to CyCMV/SIV-mediated protection against SIV replication (Fig. 3B).

The inability of the IL-15-based transcriptomic protection signature to distinguish protection outcomes following HPAI challenge indicated that unlike both RhCMV/SIV- and dd CyCMV/SIV-mediated protection against SIV replication[29,36], CyCMV/Flu-mediated protection against lethal HPAI infection did not depend upon MHC-E-restricted CD8 + T cells and the IL-15 signaling pathway. Therefore, we next assessed if the magnitude of the vaccine-induced T cell response might correlate with outcome. To this end, we examined the magnitude of influenza transgene-specific T cells immediately prior to entry into the ABSL3 for influenza challenge, which revealed that MCM surviving HPAI challenge mounted significantly higher influenza-specific CD4 + T cell responses in peripheral blood compared to MCM that succumbed (Fig. 3C). In particular, CD4 + T cells targeting the M protein in blood were significantly associated with survival. In contrast, there was no association observed with the magnitude of influenza-specific CD8 + T cells in blood prior to challenge. Based on these observations, we performed the same analysis using T cell frequencies measured in BAL, which again revealed that influenza-specific CD4 + T cells, but not CD8 + T cells, correlated with survival following challenge with HPAI (Fig. 3D). Cumulatively, these data indicate that unlike CMV vector-mediated protection against SIV replication, protection against lethal HPAI infection does not depend on CD8 + T cells, regardless of their MHC restriction, but rather on the magnitude of influenza-specific CD4 + T cell prior to infection.

## Discussion

The development of a universal influenza vaccine remains a top global health priority, but cannot be achieved via current approaches that generate strain-specific humoral immune responses. Indeed, as recently demonstrated by SARS-CoV-2, despite the ability to rapidly produce lipid nanoparticle mRNA-based vaccines, pathogen variability inevitably still yields viral variants that escape vaccine-induced neutralizing antibodies and subsequently circulate in the human population[37]. As such, novel vaccine approaches are needed that target conserved viral regions and provide heterosubtypic protection. To this end, we explored the potential of a CyCMV-vectored vaccine expressing conserved internal influenza antigens to protect against heterologous HPAI in a stringent macaque model of aerosolized influenza challenge. While murine CMV (MCMV) vectors expressing either a single influenza CD8 + T cell epitope or the entire HA protein have previously been shown to confer protection against influenza in mice[38,39], the data presented here represents, to our knowledge, the first test of CMV vectors against influenza in primates. We found that CyCMV/Flu vectors induced influenza-specific CD8+ and CD4 + T$_{EM}$, with the influenza-specific CD8 + T cells being restricted by MHC-Ia in FL CyCMV/Flu-vaccinated MCM and MHC-II or MHC-E in dd CyCMV/Flu-vaccinated MCM. Regardless of MHC-restriction, both CD8+ and CD4 + T cells from CyCMV/Flu-vaccinated MCM recognized a panel of diverse influenza isolates in vitro. As expected, based on results from CyCMV/SIV-vaccinated MCM challenged with SIV[29], CyCMV/Flu vaccination did not prevent acquisition of influenza in MCM. However, CyCMV/Flu-vaccinated MCM exhibited statistically significant protection from an otherwise lethal HPAI aerosolized challenge. This protection illustrates the potential of T$_{EM}$ responses for development of a universal influenza vaccine given that the vaccine encoded 1918 influenza M1, NP, and PB1 protein sequences and the H5N1 HPAI isolate circulated in 2004, yielding 86 years of global influenza evolution between the vaccine and heterologous challenge virus. Therefore, CMV-induced T$_{EM}$ responses targeting conserved viral antigens should be considered for inclusion in preventative approaches against pathogens such as influenza where sequence diversity in targets of antibody-mediated neutralization have stymied production of universally protective vaccines.

There exist many advantages for the CMV vaccine platform, which are not shared by other virus-based vaccine vectors. First, CMV is able to efficiently superinfect previously CMV-infected individuals despite pre-existing anti-CMV immunity[22–24,40]. Second, CMV is highly immunogenic and elicits robust memory CD4+ and CD8 + T cell immunity. Indeed, in natural HCMV infection in humans, approximately 10% of circulating memory CD4+ and CD8+ memory T cells are HCMV-specific[41]. Third, CMV induced T cell responses are highly effector differentiated, thereby equipping these T cells with the ability to mediate an immediate effector response without the need to first anamnestically expand and differentiate[22–24]. Finally, in keeping with an effector memory phenotype, CMV-induced CD4+ and CD8 + T cells are widely distributed at high frequency in various anatomical tissues, including the lungs. Although the CyCMV vaccine vectors here are spread-attenuated[30], they may still establish persistence in pulmonary myeloid lineage cells, thereby recruiting CyCMV/Flu-specific T cells to the lung. Based on these intriguing properties, and the unique protection against SIV replication observed in RhCMV/SIV-vaccinated RM, clinical trials are currently underway to test the safety and immunogenicity of a HCMV vaccine vector for HIV[26], Thus, the HCMV vaccine platform could be utilized to clinically test the approach presented here for a universal influenza vaccine.

Correlates analysis revealed that the previously identified IL-15 transcriptomic signature did not predict protection from lethal HPAI infection and that CD8 + T cells, regardless of MHC-restriction, did not associate with protection. Rather, the magnitude of the CyCMV-induced, influenza-specific CD4 + T cell response correlated to protection from HPAI-induced death. Although initially surprising given that both CyCMV/SIV- and RhCMV/SIV-mediated protection against SIV replication depends on the presence of SIV-specific, MHC-E-restricted CD8 + T cells and an IL-15-based signaling pathway[17,29,36], this observation is in line with previous reports of the importance of CD4 + T cells in influenza infection. Multiple studies have demonstrated that memory CD4 + T cells mediate heterosubtypic protection in murine models of influenza[42–45]. While influenza-specific CD4 + T cells can orchestrate an effective immune response via interactions with both B cells and CD8 + T cells, they also demonstrate direct antiviral activity mediated via production of perforin and IFN-γ[42,46]. Furthermore, the frequency of pre-existing NP- and M-specific CD4 + T cells correlated with less severe disease in human clinical studies following infection with previously unencountered influenza strains[10]. Thus, our correlate of protection based on the magnitude of pre-infection frequencies of vaccine-induced IFN-γ + CD4 + T cells is in line with previous studies, and indicates that CMV vector-mediated protection against influenza could be improved by refining the vaccine to optimize the priming of influenza-specific CD4 + T cells. However, the mechanism of protection mediated by influenza-specific CD4 + T cells following CMV vector vaccination remains undefined. We previously demonstrated that CyCMV vaccine vectors elicit little to no antibody responses against antigenic inserts present in the vaccine vector[29]. Therefore, it is unlikely that the CyCMV/Flu-elicited CD4 + T cells participated in priming a pre-existing antibody response targeting the internal viral proteins utilized as vaccine targets. Given that influenza-specific CD4 + T cells can exhibit direct antiviral activity[42,46], we hypothesize that CyCMV/Flu-elicited CD4 + T cells mediated protection by directly inhibiting viral replication. IL-15 has previously been identified to support the generation of superior, lung-resident antiviral CD4 + T cells with the ability to protect against lethal influenza infection in mice[47].

Since IL-15 plays a critical role in the development of effector differentiated CD4 + T cells, including cytotoxic CD4 + T cells[48], it is interesting that the IL-15-based transcriptomic signature that correlated with SIV protection mediated by MHC-E-restricted CD8 + T cells did not predict the protection against lethal HPAI observed here in the current study. It is therefore possible that IL-15 signaling was not a limiting factor for the function of the influenza-specific CD4 + T cells here as for the MHC-E-restricted CD8 + T cells in SIV protection[29,36]. Subsequent statistically powered CyCMV/Flu vaccine-based HPAI challenge studies in MCM are required to determine the mechanism of protection mediated by influenza-specific CD4 + T cells.

Although MHC-E-restricted CD8 + T cells were not required for protection in the MCM model of influenza, the data presented here furthers our understanding of the epitope binding capacity of MHC-E by demonstrating that this monomorphic MHC molecule can bind a variety of influenza peptides for presentation to CD8 + T cells. Indeed, this observation is in line with two recent reports describing the ability of Qa-1, the murine orthologue of HLA-E in humans, and HLA-E to bind influenza-derived epitopes[49,50]. Of note, the HLA-E-binding influenza epitopes identified were derived from NP, which we also found can generate peptide epitopes capable of binding Mafa-E, the HLA-E orthologue in MCM, and activating MHC-E-restricted CD8 + T cells. Interestingly, Jost et al. further demonstrate that the NP-derived, HLA-E-binding peptide epitopes are targeted by antigen-specific NK cells, suggesting that such antigen-specific NK cell responses could contribute to the antiviral response against influenza[49]. Whether such responses could be elicited by CyCMV/Flu vaccine vectors remains unknown, and subsequent studies are required to fully investigate for the presence of memory NK cells following vaccination with CMV vectors. However, given the myriad of interactions documented to occur between CMV and NK cells[21,51,52], it is possible that CMV vectors may be able to prime memory NK cell responses.

Overall, we demonstrate that CyCMV vaccine vector-induced, influenza-specific CD4 + $T_{EM}$ protect MCM from an otherwise lethal HPAI infection. While it remains likely that a universal influenza vaccine will have to engender both humoral and cellular immune responses, the CMV vaccine vector platform is unique given its ability to elicit and maintain long-lived $T_{EM}$, which could pair with a separate platform to stimulate both arms of the immune system. Furthermore, given that CMV vectors are now in clinical trials for HIV[26], there exists a direct pathway forward for testing influenza-specific HCMV vectors in humans. Further such studies are required to determine if CMV-induced $T_{EM}$ can contribute to the development of a universal influenza vaccine.

## Methods
### Production of CyCMV/influenza vaccines
The CyCMV vector constructs used in this study were based on the published FL-CyCMV (strain 31908) bacterial artificial chromosome (BAC) and were generated by en passant homologous recombination[28,29]. The codon optimized DNA sequences of the Influenza A virus (IAV) (A/Brevig Mission/1/1918(H1N1) derived NP (AY744935), M1 (AY130766) and PB1 (DQ208310) genes were synthesized by GENEWIZ (Azenta Life Sciences). No M2 sequences were included. Within each DNA sequence, a 50 bp stretch was duplicated and the two homologous sequences were separated by an I-SceI restriction site and an aminoglycoside 3′-phosphotransferase (Kanamycin resistance, KanR) selection cassette. The vaccine inserts were amplified with recombination primers carrying a 50 bp overhang homologous to the upstream and downstream region of the Cy110 (UL82, pp71) ORF. Homologous recombinations were performed in *E.coli* strain GS1783 which can be used to express the λ phage derived Red recombination genes after heat shock induction and recombination resulted in the substitution of the Cy110 ORF with the vaccine inserts using endogenous viral regulatory elements to drive transgene expression[25,29,30]. Successful recombinants were analyzed by XmaI restriction digest and Sanger sequencing across the altered genomic locus to ensure genome integrity. The KanR cassette was removed from the vaccine insert by inducing a DNA double strand break in the BAC at the I-SceI- recognition site through the arabinose induced expression of I-Sce I in *E.coli* strain GS1783. Simultaneous expression of the Red recombination genes through heat shock induction resulted in homologous recombination of the introduced 50 bp homologous sequences in the DNA insert leading to seamless removal of the selection maker from the BAC. Final clones were once more analyzed by XmaI restriction digest and Sanger sequencing across the altered genomic locus and next generation sequencing of the full BAC was performed to exclude off-target mutations.

Primary rhesus fibroblasts (RFs) were transfected with the final BAC constructs using Lipofectamine 3000 (Invitrogen) to reconstitute the viral vectors. Initial seed stocks were expanded into eight confluent T-175 flasks of RFs, and cells and supernatants were harvested at full CPE and frozen at −80 °C overnight to release cell associated virus. Vector purification was performed by clarifying the supernatants by centrifugation, first at 2000 x $g$ for 10 min at 4 °C and subsequently at 7500 x $g$ for 15 minutes. Lastly, the CyCMV vectors were pelleted through a sorbitol cushion (20% D-sorbitol, 50 mM Tris [pH 7.4], 1 mM MgCl2) by centrifugation at 64,000 x $g$ for 1 h at 4 °C in a Beckman SW28 rotor. The purified viral stocks were resuspended in DMEM complete, aliquoted in 125 ul aliquots and stored at −80 °C until final use. Viral titers of the purified viral stocks were determined in triplicates by immunofluorescence assay (IFA) as for HCMV[53]. RFs were infected with serial dilutions of each purified vector and the cells were fixed with 100% methanol at ≤− 20 °C after 72 h. The cells were first stained with an α-RhCMV pp65b antibody 19C12.2 and subsequently with an Alexa Fluor 488-conjugated goat anti-mouse IgG secondary (Invitrogen) for 1 h at 37 °C[54]. Afterwards, the cells were washed three times with PBS and the nuclei were stained with DAPI for 20 min at room temperature. An EVOS fluorescence microscope (Life Technologies) was used to acquire images of the titration plates which were processed and analyzed using the ImageJ software. The ratio of infected to uninfected cells was used as a measure to backcalculate the number of focus forming units per milliliter (FFU/ml) in each purified CyCMV vaccine vector stock.

### Mauritian cynomolgus macaques
Mauritian-origin cynomolgus macaques (MCM) were purchased from a commercial vendor following serological testing confirming the absence of antibodies to influenza A and B viruses (Supplementary Table 1). MCM were split randomly into three experimental groups based on sex to yield equal numbers of male and female MCM in each group. All MCM were less than four years of age. MCM were moved to Tulane National Primate Research Center (TNPRC) for the vaccine phase of the project, which was reviewed and approved by the institutional Animal Care and Use Committee of Tulane University. Animals were cared for in accordance with the Guide for the Care and Use of Laboratory Animals. Procedures for handling and ABSL2 containment of animals were approved by the Tulane University Institutional Biosafety Committee. The TNPRC is fully accredited by the Association for the Assessment and Accreditation of Laboratory Animal Care International. Challenge of the MCM was performed at the University of Pittsburgh, which is fully accredited by the Association for Assessment and Accreditation of Laboratory Animal Care International. Prior to challenge, the work with animals described in this report was approved by the University of Pittsburgh's Institutional Animal Care & Use Committee (IACUC). CyCMV vectors were delivered subcutaneously at $1 \times 10^7$ PFU per vector.

### H5N1 aerosol challenge
The virus (H5N1 A/Vietnam/1203/2004) used in this study was generated via reverse genetics[27,55] by Dr. S. Mark Tompkins at the Department of Infectious Diseases, University of Georgia, who provided the original reverse genetics stock which has been passaged twice in eggs. Under BSL3 conditions at the University of Pittsburgh Regional Biocontainment Laboratory, the virus was inoculated into 10-11 day-old embryonated specific pathogen free chicken eggs to generate a stock for use in these studies. Allantoic fluid was recovered at 24 h from inoculated eggs and clarified by centrifugation. Aliquots were then stored at −80 °C and virus titer was determined by plaque assay in MDCK cells. For aerosol challenges, virus was diluted in DMEM containing bovine serum albumin, HEPES buffer, and penicillin/streptomycin[56]. The virus stock was sequenced to confirm identity with the published sequence for A/Vietnam/1203/2004[27]. Work done with this virus described in this report was approved by the University of Pittsburgh's Institutional Biosafety Committee (IBC), Biohazards Committee, and Department of Environmental Health & Safety prior to initiation of experiments.

Aerosol exposures were performed using a Aerogen Solo (Aerogen) vibrating mesh nebulizer controlled by the Aero3G aerosol management platform (Biaera Technologies)[27,55,56]. MCM were anesthetized with 6 mg/kg Telazol (Tiletamine HCl / Zolazepam HCl) and transported to the Aerobiology suite using a mobile transport cart. The MCM was then transferred from the cart into a class III biological safety cabinet and the macaque's head was placed inside a head-only exposure chamber. For accumulated tidal volume (ATV) exposures, Jacketed External Telemetry Respiratory Inductive Plethysmography (JET-RIP; Data Sciences International) belts were placed around the upper abdomen and chest of the macaque and calibrated to a Hans Rudolph pneumotach (Shawnee, KS) in Ponemah 5.4 software DSI (Data Sciences International). Once the belts were calibrated, virus was put into the nebulizer at a concentration of $5 \times 10^6$ pfu and the exposure started. During the exposure, tidal volume data was transferred from

Ponemah to the Biaera software. Exposures were terminated when the macaques had breathed in a total of 6 liters of virus-laden air to achieve a target inhaled dose of $2 \times 10^5$ pfu[57]. If the belts could not be properly calibrated, exposures were time-calculated using minute volumes collected by a head-out plethysmography three days prior to challenge and the spray factor for H5N1 based on past performance. Exposures were dynamic, set to a total airflow of 16 liter per minute (lpm) of air into and out of the chamber (one complete air change every 2 min)[57]. Inhaled dose was determined by plaque assay on samples collected in an all-glass impinger (AGI; Ace Glass) attached to the chamber and at operated at 6 lpm, -6 to -15 psi throughout the exposure. Particle size was measured using an Aerodynamic Particle Sizer (TSI, Shoreview, MN); one sample was collected for 30 s at 5 min after the exposure had started. Following the exposure, the MCM was subjected to a 5 min air wash after which the macaque was removed from the cabinet and transported back to its cage and observed until fully recovered from anesthesia. Virus concentration in nebulizer was assessed by plaque assay to evaluate aerosol performance relative to previous aerosol exposures; inhaled dose was calculated as the aerosol concentration of the virus determined from the AGI multiplied by the accumulated volume of inhaled air (6 liters).

## Clinical scoring of chest radiographs and humane endpoints

Clinical signs were recorded at least twice daily and given an objective score to ensure that severely ill or moribund animals were identified quickly. The scoring system included body temperature (especially hypothermia), clinical appearance, and respiratory signs (cageside observation as well as plethysmography data and SPO2 readings) as follows: Temperature: normal range 36 to $39.5\,°C = 0$; Elevated > $39.5\,°C = 1$; Hypothermia $34-36\,°C = 2$; Severe Hypothermia < $34\,°C = 3$. Clinical Appearance: Normal=0; Lethargic, huddled=1; Moves only when prodded=3. Respiratory Symptoms: None=0; Nasal discharge=1; Increased respiratory rate and effort=2; Respiratory distress (defined as taking shallow breaths at twice the resting rate)=3. A combined score of 4 or a score of 3 in any one category required increased direct observation to every 4–6 h. The telemetry system was set to send alerts if body temperature dropped below $36\,°C$, which triggered an immediate cage side observation to determine the cause. Macaques that reached a clinical score of 6, were severely hypothermic (body temperature <$34\,°C$ for ≥4 h), or were found to be unresponsive were promptly euthanized. Ventro-dorsal radiographs of sedated macaques were taken using an SRI portable radiographic unit with digital radiograph processing using a Fujifilm processor. Radiographs were scored by two radiologists blinded to the vaccine and infection state of the macaques using a scale of 0–3 with each lung divided into three fields, for a total scoring range of 0–18[58].

## Telemetry

DSI PhysioTel Digital radiotelemetry transmitter (DSI Model No. M00) capable of continuously recording body temperature and activity were implanted abdominally in all the macaques used in this study. Macaques were allowed to heal for at least 14 days prior to transfer into the RBL and challenge. Implants were turned on 4-5 days prior to aerosol challenge to collected baseline data for modeling. Data was transmitted from the implant to TRX-1 receivers mounted in the room connected via a Communications Link Controller (CLC) to a computer running Ponemah v6.5 (DSI) software. Data collected from Ponemah was exported as 15 min averages into Excel files which were subsequently analyzed in MatLab 2019a as previously described[59,60]. Pre-exposure baseline data was modeled using auto-regressive integrated moving average (ARIMA) to forecast body temperature after challenge, assuming no significant change in temperature. Residual temperatures were calculated as actual minus predicted temperatures. Significant elevations or decreased in temperature were determined by upper and lower residual limits calculated as the product of 3 times the square root of the residual sum of squares from the baseline data. Maximum deviation in temperature (Max $\Delta T$) was the highest residual difference between actual and predicted body temperature after challenge. Fever duration was calculated in hours, dividing the number of significant elevations by 4. Fever severity was calculated as the sum of all significant elevations in body temperature after challenge, divided by 4 to get fever-hours. Average elevation was calculated by dividing fever severity by fever duration.

## Plaque assay

Infectious virus in tissue homogenates, BAL, nasal/oral, and aerosol samples were determined by standard plaque assay in MDCK cells. Prior to plaque assays, MDCK cells in DMEM-10 media were put into six-well plates and grown until they were 70–80% confluent. Snap-frozen tissues were homogenized in media containing FBS using an Omni tissue homogenizer (Omni International). Samples were diluted serially 10-fold and inoculated onto MDCK cells (200 µl per well) for 1 hour at $37\,°C$ a plates before being overlaid with 1% agarose-containing EMEM. Plates were then incubated for 3 d at $37\,°C$, 5 d at $37\,°C$ for tissues, fixed overnight in 10% formaldehyde at room temperature, and finally stained with 0.25 % crystal violet to visualize plaques.

## Statistics & reproducibility

Mixed effect model with antigen as repeated measures was performed to compare T cell responses by different vaccination methods (FL vs dd CyCMV/Flu) and outcomes (survived vs. terminal). For the in vitro influenza challenge, t-test was conducted for comparing T cell responses of different vaccination methods under each virus incubation. Viral titer and thoracic radiographic scores were compared among different treatment-by-outcome combinations or among different timepoints, using ANOVA with Tukey-Kramer comparison, as some comparisons were not estimable using the mixed effect model due to missing data of early terminations. Kaplan-Meier with log-rank test were performed for survival. Viral load analysis was run on log 10-transformed data. Analysis was performed with SAS9.4 (PROC MIXED, PROC TTEST and PROC GLM) software. A sample size of $n = 6$ animals per group was selected based on previous studies with CMV vaccine vectors, thus no statistical method was used to predetermine sample size. Animals were assigned randomized into three experimental groups based on sex to yield equal numbers of males and females in each group. No data were excluded from the analyses. Investigators performing the influenza challenges were blinded to the vaccine status of animals until after completion of the study.

## Immunological assays

PBMC was isolated from EDTA-treated whole blood using Ficoll-Paque (GE Healthcare) density centrifugation[52,61–63]. Cells were resuspended in RPMI 1640 containing 10% FBS (R10; Hyclone Laboratories, Logan, UT). CD4+ and CD8 + T cells specific for influenza were detected as described for other pathogens using flow-cytometric intracellular cytokine analysis[29,64–67]. Sequential 15-mer peptides (Genscript) that overlap by 11 amino acids comprising the sequence of Influenza A virus (A/BrevigMission/1/1918(H1N1)) Matrix1 (M1), Nucleoprotein (NP), or Polymerase-Basic 1 (PB1) proteins were combined with PBMC or mononuclear cells from BAL and co-stimulatory antibodies anti-CD28 and anti-CD49d. Cells were combined with peptide antigen and incubated for 1 h at $37\,°C$ 5% $CO_2$ before the addition of Brefeldin A and an additional eight-hour incubation. After incubation, cells were chilled at $4\,°C$ overnight. Co-stimulation without antigen served as a negative control. Cells were then stained with fluorochrome conjugated antibodies listed below and data was acquired on an LSRII (BD Biosciences) and analyzed using FlowJo software (BD Biosciences). CD4+ and

CD8 + T cell antigen specific responses were determined by the boolean expression of CD69 + TNFα + OR CD69 + IFNγ+ frequencies. Longitudinal analysis of responses were memory corrected using CD28 and CD95 markers to define memory populations[68]. MHC blocking ICS were performed as above except peptide stimulation was preceded by the addition of one of each the following specific inhibitors: 1) the pan anti-MHC-I mAb W6/32 (10 mg/mL), 2) the MHC-II-blocking mAb L243 (10 mg/mL), or 3) the MHC-E blocking VL9 peptide (VMAPRTLLL; 20 μM) for one hour before peptide was added[20,69]. To be considered MHC-E restricted by blocking, the individual peptide response must have been blocked by both anti-MHC-I clone W6/32 and MHC-E-binding peptide VL9, and not blocked by the anti-MHC-II clone L243. To be considered MHC-II-restricted by blocking, the individual peptide response must have been blocked by the anti-MHC-II clone L243 and not blocked by either the anti-MHC-I clone W6/32 or the MHC-E-binding peptide VL9. To be considered MHC-Ia-restricted by blocking, the individual peptide response must have been blocked by the anti-MHC-I clone W6/32 and not blocked by either the MHC-E-binding peptide VL9 or the anti-MHC-II clone L243. For influenza subtype recognition assays, PBMC were incubated with beta-propiolactone (BPL)-inactivated influenza virus in conditions matching previously described peptide-stimulations and then intracellular cytokine stained as described above. The following reagents were obtained through the International Reagent Resource, Influenza Division, WHO Collaborating Center for Surveillance, Epidemiology and Control of Influenza, Centers for Disease Control and Prevention, Atlanta, GA, USA: BPL-Inactivated Influenza A Virus, A/Vietnam/1203/2004 (H5N1), FR-736, BPL-Inactivated Influenza A Virus, A/Bangladesh/3002/2015 (H1N1)pdm09, FR-1458, BPL-Inactivated Influenza A Virus, A/Shanghai/2/2013 (H7N9), FR-1390, BPL-Inactivated Influenza A Virus, A/Sichuan/26221/2014 (H5N6), FR-1433, BPL-Inactivated Influenza A Virus, A/Hong Kong/33982/2009 (H9N2), FR-775, BPL-Inactivated Influenza A Virus, A/Ohio/02/2012 (H3N2), FR-1144, BPL-Inactivated Influenza A Virus, A/Anhui/1/2013 (H7N9), FR-1283, BPL-Inactivated Influenza A Virus, A/pheasant/New Jersey/1355/1998 (H5N2), FR-912, and BPL-Inactivated Influenza A Virus, A/mallard/Netherlands/12/2000 (H7N7), FR-914.

## Antibodies

To define the memory vs. naïve subsets, the following antibodies were used: SP34-2 (CD3; BUV395, BD Biosciences), SK-1 (CD8; BUV737; BD Biosciences), G043H7 (CCR7; Biotin; BioLegend), Streptavidin (BV421; BioLegend), L243 (HLA-DR; BV510; BioLegend), L200 (CD4; BV786; Fisher Scientific), B57 (Ki67; FITC; BD Biosciences), DX2 (CD95; PE; BioLegend), CD28.2 (CD28; PE/Dazzle 594; BioLegend), CH/4 (CD69; PE-Cy5.5; Life Technologies), 3A9 (CCR5; APC; BD Biosciences), 2H7 (CD20; APC-Fire 750; BioLegend). For T cell response and recognition assays, the following antibodies were used: CD28.2 (CD28; Pure; Life Technologies), 9F10 (CD49d; Pure; Life Technologies), SP34-2 (CD3; Pacific Blue; BD Biosciences), L200 (CD4; BV510; eBiosciences (PE) Biolegend (PE/Dazzle 594)), B57 (Ki67; FITC; BD Bioscience), Mab11 (TNFα; PE, FITC; BioLegend), FN50 (CD69; PE, PE/Dazzle 594; BioLegend), SK-1 (CD8a; PerCP-eFluor 710; Life Tech) and B27 (IFNγ; APC; BioLegend).

## Influenza phylogenetic analysis

The nucleotide sequences from the following influenza strains were aligned using Clustal Omega[70], aligning each segment independently, followed by neighbor-joining phylogenetic tree construction using Geneious software: A/Alabama/01/2020 (H1N1), A/California/03/2019 (H1N1), A/Aichi/2/68 (H3N2), A/Hong Kong/01/1968 (H3N2), A/Cambodia/X0123311/2013 (H5N1), A/Alabama/01/2010 (H1N1), A/Anhui/1/2005 (H5N1), A/Aalborg/INS132/2009 (H1N1), A/Japan/305/1957 (H2N2), A/Guiyang/1/1957 (H2N2), A/Albany/20/1957 (H2N2), A/Albany/10/1968 (H2N2), and A/Brevig Mission/1/1918 (H1N1).

## RNAseq

Whole blood was collected from MCM into PAXgene RNA tubes (PreAnalytiX) according to the manufacturer's instructions. RNA was isolated using RNAdvance Blood Kit (Beckman) following the manufacturer's instructions. mRNAseq libraries were constructed and sequenced using Illumina TruSeq Stranded mRNA HT kit following the manufacturer's recommended protocol. Libraries were sequenced on an Illumina NextSeq500 sequencer using Illumina NextSeq 500/550 High Output v2 kits (150 cycles) following the manufacturer's protocol for sample handling and loading[36]. Raw sequencing reads were demultiplexed with bcl2fastq. Residual adapters and low quality bases were then removed with Trim Galore, a wrapper for cutadapt, followed by globin read removal with bowtie2 v2.4.2[71]. Filtered reads were mapped and quantified to the RM genome (Mmul10, Ensembl v100) with STAR v2.7.5[72]. Gene counts were imported into the R statistical software for subsequent analyses. We first removed lowly expressed genes, then normalized the filtered counts using TMM normalization[73] followed by voom transformation[74]. Differential expression analyses were performed with limma and EdgeR, testing for changes in expression within each group (protected and nonprotected), at each time point post-vaccination relative to baseline (adj $p$ value < 0.05 using Benjamini & Hochberg method, absolute $\log_2$ fold change >1.5). A previous set of 122 genes had been identified as important for protection in CMV vaccine that conveyed protection against SIV[29]. We plot the LFC of these genes in using heatmap.2 clustered with the same methods outlined above.

## Reporting summary

Further information on research design is available in the Nature Portfolio Reporting Summary linked to this article.

## Data availability

The bulk sequencing data has been deposited in GEO (GSE268204) and analysis code has been deposited in a Github repository and can be accessed via this link: https://github.com/galelab/Sacha_CyCMV-FLU_2024. The code for auto-regressive integrated moving average (ARIMA) for body temperature is available at https://github.com/ReedLabatPitt/Reed-Lab-Code-Library. Source data are provided with this paper.

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

## Acknowledgements

This work was supported by the Bill & Melinda Gates Foundation Grand Challenges grant OPP1213553 to JBS, and R01 AI40888 to J.B.S. and S.G.H., and with support from P51 OD011092 from the NIH Office of the Director to the Oregon National Primate Research Center (ONPRC P51 Core grant, which supports salary for J.B.S., S.G.H., K.F., and L.J.P.). The RNA sequencing assay was performed by the WaNPRC Seattle Genomics service core. Funding for Seattle Genomics is supported in part by the National Institutes of Health, Office of the Director P51OD010425 (M.G.J.). We thank members of the ONPRC Virology Core for production of CMV vectors. The following reagents were obtained through the International Reagent Resource, Influenza Division, WHO Collaborating Center for Surveillance, Epidemiology and Control of Influenza, Centers for Disease Control and Prevention, Atlanta, GA, USA: BPL-Inactivated Influenza A Virus, A/Vietnam/1203/2004 (H5N1), FR-736, BPL-Inactivated Influenza A Virus, A/Bangladesh/3002/2015 (H1N1)pdm09, FR-1458, BPL-Inactivated Influenza A Virus, A/Shanghai/2/2013 (H7N9), FR-1390, BPL-Inactivated Influenza A Virus, A/Sichuan/26221/2014 (H5N6), FR-1433, BPL-Inactivated Influenza A Virus, A/Hong Kong/33982/2009 (H9N2), FR-775, BPL-Inactivated Influenza A Virus, A/Ohio/02/2012 (H3N2), FR-1144, BPL-Inactivated Influenza A Virus, A/Anhui/1/2013 (H7N9), FR-1283, BPL-Inactivated Influenza A Virus, A/pheasant/New Jersey/1355/1998 (H5N2), FR-912, and BPL-Inactivated Influenza A Virus, A/mallard/Netherlands/12/2000 (H7N7), FR-914.

## Author contributions

DM and MT contributed equally as co-first authors. DM, CRP, and HT constructed, validated, and produced the CyCMV vaccine vectors overseen by KF and LJP. RMG, DWM, CMH, JR, and AS measured T cell responses ex vivo overseen by SGH. TP, PPA, FS, and JPD performed animal work during the vaccine phase, overseen by NJM. PC, MW, YY, MM, CW, GY, AL, and JL performed animal work and analyzed data during the challenge phase overseen by SB-B and DSR. LT analyzed blinded radiographs. YY and LG performed statistical analyses. JT-G and LSW generated and performed transcriptomic analyses under the supervision of MGJr. JBS conceived of the project, secured funding, and supervised the study. JBS wrote the manuscript with help from SGH, SB-B, and DSR. All authors read and commented on the manuscript.

## Competing interests

O.H.S.U. and D.M., K.F., L.J.P., S.G.H., and J.B.S. have a significant financial interest in Vir Biotechnology, Inc., a company that may have a financial interest in the results of this research and technology. This potential individual and institutional conflict of interest has been reviewed and managed by O.H.S.U. The remaining authors declare no competing interests.

## Additional information

**Daniel Malouli**[1,8], **Meenakshi Tiwary**[1,8], **Roxanne M. Gilbride**[1], **David W. Morrow** ®[1], **Colette M. Hughes**[1], **Andrea Selseth** ®[1], **Toni Penney**[2], **Priscila Castanha**[3], **Megan Wallace**[3], **Yulia Yeung**[3], **Morgan Midgett**[4], **Connor Williams**[3], **Jason Reed**[1], **Yun Yu**[1], **Lina Gao** ®[1], **Gabin Yun**[5], **Luke Treaster**[5], **Amanda Laughlin** ®[4], **Jeneveve Lundy**[4], **Jennifer Tisoncik-Go** ®[6], **Leanne S. Whitmore** ®[6], **Pyone P. Aye** ®[2], **Faith Schiro**[2], **Jason P. Dufour**[2], **Courtney R. Papen**[1], **Husam Taher**[1], **Louis J. Picker**[1], **Klaus Früh** ®[1], **Michael Gale Jr** ®[6,7], **Nicholas J. Maness** ®[2], **Scott G. Hansen**[1], **Simon Barratt-Boyes** ®[3,9] ✉, **Douglas S. Reed** ®[4,9] ✉ & **Jonah B. Sacha** ®[1,9] ✉

[1]Oregon National Primate Research Center, Vaccine & Gene Therapy Institute, Oregon Health & Science University, Beaverton, OR, USA. [2]Tulane National Primate Research Center, Tulane University, New Orleans, LA, USA. [3]Department of Infectious Diseases and Microbiology, Pittsburgh, PA, USA. [4]Center for Vaccine Research, Pittsburgh, PA, USA. [5]Department of Diagnostic Radiology, University of Pittsburgh, Pittsburgh, PA, USA. [6]Center for Innate Immunity and Immune Disease, University of Washington, Seattle, WA, USA. [7]Washington National Primate Research Center, Seattle, WA 98195, USA. [8]These authors contributed equally: Daniel Malouli, Meenakshi Tiwary. [9]These authors jointly supervised this work: Simon Barratt-Boyes, Douglas S. Reed, Jonah B. Sacha. ✉e-mail: smbb@pitt.edu; dsreed@pitt.edu; sacha@ohsu.edu

