## [Peer Review File · Nature Communications]

Cytomegalovirus vaccine vector-induced effector memory CD4 + T cells protect cynomolgus macaques from lethal aerosolized heterologous avian influenza challengeREVIEWER COMMENTS

Reviewer #1 (Remarks to the Author):

The authors examined the immune responses induced by vaccination with two CMV vector vaccines in cynomolgus macaques. They selected influenza M, NP, and PB1 proteins to induce heterosubtypic immune responses against influenza A virus. Challenge infection with H5N1 highly pathogenic avian influenza virus showed partial protection in vaccinated macaques.

1. In Fig. 1D-F and Fig. 3C and D, please indicate the days on which samples were collected from monkeys.
2. In Fig. 2, the dd CyCMV/Flu vaccine resulted in a better efficacy in a survival rate and viral titers in H5N1 challenge infection. What made the difference?
3. In lines 212 and 219, 'Figure 2B' and 'Figure 2C' should be Figure 2C and E, respectively.
4. Fig. 3B requires explanation in detail. Gene names including IL-15 related genes should be added.
5. In Fig. 3D, how did CD4+ T cells contribute for survival? What are the percentages of TCM and TEM responding to each antigen?
6. The relationship between IL-15-based transcription and CD4+ T cells should be examined and discussed.
7. Since many humans have been infected with CMV and may have immunity specific against CMV, is the CMV vector effective in humans preinfected with CMV and how is the CMV vector applied for humans?

Reviewer #2 (Remarks to the Author):

The manuscript described research to develop a T cell-targeted universal influenza vaccine by generating CMV vaccine vectors (FL CyCMV and dd CyCMV) expressing influenza internal NP, M, and PB1 proteins. Vaccination studies in Mauritian cynomolgus macaques (MCM) with the resulting vaccine vectors induced lung-resident influenza-specific effector memory CD4 T cells conferring protection against an HPAI challenge. The study determined new targets, methodology, and correlates of influenza cross-protection for future universal influenza vaccine development. The publication of this study is of interest to many readers. In reviewing the manuscript, the reviewer identified some comments/questions that the reviewer believes the paper can be further improved by addressing these concerns.

Comments:

1. Different vaccine vectors, like rAd vectors, have been studied to induce enhanced immune responses. Compared to other recombinant viral vectors, what is the advantage of the CMV vaccine vectors in inducing protective influenza T cell immunity?

2. CyCMV vaccination did not induce meaningful lung resident memory CD8 T (CD8 TRM) cells. However, influenza-specific lung CD8 TRM cells can be solely induced by local antigen presentation, like influenza infection or intranasal vaccination, and have been identified to correlate with influenza heterosubtypic protection by providing rapid recall responses. It is interesting to know whether CyCMV subcutaneous vaccination can lead to significant local antigen expression and presentation, specifically in the lung. Is it possible that low levels of local antigen expression (in lungs) in CyCCMV vaccination caused the low CD8 TRM cells seen in the study?

3. CyCMV flu vaccines did not induce the protection signature IL-15 response pathway gene expression in CyCMV/SIV vaccination. What is the immunological difference between SIV and flu antigens in generating such different immune responses?

Reviewer #3 (Remarks to the Author):

Malouli and Tiwary et al describe the development and testing of a CMV vector-based influenza vaccine in NHPs, designed to induce protective T cell responses against H5N1 challenge. This is an interesting (and now particularly timely) study of novel approaches to induce broad protection against influenza strains not currently circulating in human populations. The vaccines elicit both CD4 and CD8 T cell responses against the three antigens of interest, and provide partial protection against ARDS/severe disease following aerosol challenge. Interestingly, despite previous studies of CMV vector vaccines against SIV and Mtb, the best correlate of protection identified in this study was CD4, rather than CD8, T cell responses.

The major question that remains unresolved is the potential mechanism by which CD4 T cells may contribute to protection from severe disease. This could include the development of cytotoxic CD4 cells, CD4-mediated B cell help driving binding antibody production (unlikely given Tem phenotype, but possible), or antiviral cytokine production. In the current study, the authors should provide further clarity on which of these functions may be relevant in this context - this could involve assessing binding antibody titres to the vaccine antigens, or further characterisation of the antigen-specific CD4 T cells.

Minor comments:

- Representative flow data and gating strategy should be provided in supplemental materials
- A reference or supplemental information should be provided for the description of the CyCMV SIV vaccine experiments described on lines 130-133
- There was a pronounced spike in CD4 T cell responses at day 56 post-vaccination (Fig 1B) - was this expected/is this a feature of the CMV vector vaccines?

Response to Reviewers:

Author responses in blue.

Reviewer #1 (Remarks to the Author):

The authors examined the immune responses induced by vaccination with two CMV vector vaccines in cynomolgus macaques. They selected influenza M, NP, and PB1 proteins to induce heterosubtypic immune responses against influenza A virus. Challenge infection with H5N1 highly pathogenic avian influenza virus showed partial protection in vaccinated macaques.

1. In Fig. 1D-F and Fig. 3C and D, please indicate the days on which samples were collected from monkeys.

We now list the days post vaccination that PBMC samples were used for these assays in the legends for Figure 1 and 3. These additions can be found on lines 464-465, 472, and 476-77.

2. In Fig. 2, the dd CyCMV/Flu vaccine resulted in a better efficacy in a survival rate and viral titers in H5N1 challenge infection. What made the difference?

While the dd CyCMV/Flu vaccine did result in increased survival, the current study was not appropriately powered to support efficacy comparison between the two vectors. In particular, the death of one vaccinated animal in the FL CyCMV/Flu vaccine group during the vaccine phase further precludes a direct comparison of the two groups. However, this result is a potential hint that real differences in the efficacy of the two vectors may exist. Given the intriguing results of this study, subsequent studies with appropriate statistical power to investigate for potential efficacy differences between FL and dd CMV vectors are being designed.

3. In lines 212 and 219, 'Figure 2B' and 'Figure 2C' should be Figure 2C and E, respectively.

We thank reviewer 1 for catching this mistake. To keep the text in the same logical flow, we have updated the panel order on Figure 2 to now match the text as written.

4. Fig. 3B requires explanation in detail. Gene names including IL-15 related genes should be added.

As requested, we now include a more detailed explanation of this figure on lines 271-275 and list the gene names in a new supplemental table 2.

5. In Fig. 3D, how did CD4+ T cells contribute for survival? What are the percentages of TCM and T EM responding to each antigen?

This point of how CD4+ T cells might contribute to survival was also raised by Reviewer 3 below. Therefore, we now discuss the potential mechanisms by which these cells might mediate their antiviral efficacy on lines 375-391. While we cannot explicitly determine the mechanism, this is a high priority experimental goal as this would facilitate the ability to increase the protective capability of this vaccine approach. As such, we are planning follow up experiments aimed at elucidating exactly how CMV-engendered CD4+ T cells contribute to survival post challenge with HPAI. The percentages of EM T cells are shown in Figure 1D for NP. Based on our previous experience measuring the T cell response against various antigenic inserts, we do not expect these values to differ significantly based on the antigen being recognized.

6. The relationship between IL-15-based transcription and CD4+ T cells should be examined and discussed.

As requested, we now include expand on the potential relationship between IL-15 and CD4+ T cells and how this might impact vaccine-mediated protection from HPAI. This discussion can be found on lines 382-388

7. Since many humans have been infected with CMV and may have immunity specific against CMV, is the CMV vector effective in humans preinfected with CMV and how is the CMV vector applied for humans?

One of the key features of the CMV-based vector platform is the unique ability to persistently superinfect CMV+ subjects, and in the superinfection process, induce immune responses to exogenous vaccine inserts present in the superinfecting CMV vector. Indeed, RhCMV and CyCMV infection is ubiquitous in all colony raised rhesus and cynomolgus macaques, respectively, and the vast majority of CMV vector-vaccinated animals used in all of our previous vaccine studies (showing efficacy in SIV, TB, and malaria) were naturally CMV infected and were protected by the immunogenicity resulting from vector superinfection. Humans have been shown to be successively infected by different HCMV strains, but whether the HCMV vector will work similarly as RhCMV or CyCMV vectors will need to be validated in clinical trials, which are ongoing. We now discuss superinfection on lines 341-343.

Reviewer #2 (Remarks to the Author):

The manuscript described research to develop a T cell-targeted universal influenza vaccine by generating CMV vaccine vectors (FL CyCMV and dd CyCMV) expressing influenza internal NP, M, and PB1 proteins. Vaccination studies in Mauritian cynomolgus macaques (MCM) with the resulting vaccine vectors induced lung-resident influenza-specific effector memory CD4 T cells conferring protection against an HPAI challenge. The study determined new targets, methodology, and correlates of influenza cross-protection for future universal influenza vaccine development. The publication of this study is of interest to many readers. In reviewing the manuscript, the reviewer identified some comments/questions that the reviewer believes the paper can be further improved by addressing these concerns.

We thank reviewer two for their overall positive assessment of our manuscript.

Comments:

1. Different vaccine vectors, like rAd vectors, have been studied to induce enhanced immune responses. Compared to other recombinant viral vectors, what is the advantage of the CMV vaccine vectors in inducing protective influenza T cell immunity?

There are multiple advantages of the CMV vector platform, including the following: 1) ability to superinfect CMV+ individuals, 2) priming of high frequency CD4+ and CD8+ T cells targeting antigenic inserts, and 3) effector memory phenotype of CMV-induced T cells yields T cells at portals of pathogen entry, such as the lung. We now include a discussion of these points on lines 341-354.

2. CyCMV vaccination did not induce meaningful lung resident memory CD8 T (CD8 TRM) cells. However, influenza-specific lung CD8 TRM cells can be solely induced by local antigen presentation, like influenza infection or intranasal vaccination, and have been identified to correlate with influenza heterosubtypic protection by providing rapid recall responses. It is interesting to know whether CyCMV subcutaneous vaccination can lead to significant local antigen expression and presentation, specifically in the lung. Is it possible that low levels of local antigen expression (in lungs) in CyCMV vaccination caused the low CD8 TRM cells seen in the study?

Yes, one of the drivers for our selection of the CMV vaccine vector for influenza is the previously-noted ability of CMV-engendered effector memory T cells to accumulate in the lungs. Whether this is due to chronic antigen stimulation within the lung itself, or migration of effector memory T cells into pulmonary tissue is not known.

3. CyCMV flu vaccines did not induce the protection signature IL-15 response pathway gene expression in CyCMV/SIV vaccination. What is the immunological difference between SIV and flu antigens in generating such different immune responses?

We apologize for the confusion around this point. It is not that the SIV protection signature was not present at all, it was that in those animals where the SIV protective transcriptomic signature was present, the signature did not correlate with protection against HPAI. Therefore, it is not that there exist immunological differences in the SIV versus influenza antigen, but rather that the previously described SIV protection signature was not predictive in this current study against lethal influenza. We now attempt to clarify this point on line 278. Subsequent studies with larger group sizes will be required to define a HPAI protection-associated protection signature.

Reviewer #3 (Remarks to the Author):

Malouli and Tiwary et al describe the development and testing of a CMV vector-based influenza vaccine in NHPs, designed to induce protective T cell responses against H5N1 challenge. This is an interesting (and now particularly timely) study of novel approaches to induce broad protection against influenza strains not currently circulating in human populations. The vaccines elicit both CD4 and CD8 T cell responses against the three antigens of interest, and provide partial protection against ARDS/severe disease following aerosol challenge. Interestingly, despite previous studies of CMV vector vaccines against SIV and Mtb, the best correlate of protection identified in this study was CD4, rather than CD8, T cell responses.

The major question that remains unresolved is the potential mechanism by which CD4 T cells may contribute to protection from severe disease. This could include the development of cytotoxic CD4 cells, CD4-mediated B cell help driving binding antibody production (unlikely given Tem phenotype, but possible), or antiviral cytokine production. In the current study, the authors should provide further clarity on which of these functions may be relevant in this context - this could involve assessing binding antibody titres to the vaccine antigens, or further characterization of the antigen-specific CD4 T cells.

We thank Reviewer 3 for their overall assessment of our manuscript. We now include a more detailed description of the potential CD4 functions that could contribute to the mechanism of protection. This can be found on lines 375-391. We acknowledge that additional mechanistic studies will be required to establish the protective function of the CD4+ T cells induced by CyCMV in this model.

Minor comments:

- Representative flow data and gating strategy should be provided in supplemental materials

We now include a new Supplemental figure 2, which shows representative flow data along with our gating strategy.

- A reference or supplemental information should be provided for the description of the CyCMV SIV vaccine experiments described on lines 130-133

We apologize for the confusion. The reference for these experiments is reference 29 which was cited on the previous sentence. To clarify, we now directly cite this manuscript again to direct the reader to the published manuscript.

- There was a pronounced spike in CD4 T cell responses at day 56 post-vaccination (Fig 1B) - was this expected/is this a feature of the CMV vector vaccines?

T cell responses to CMV and its antigenic inserts, while maintained at high frequency, do fluctuate over time. We have observed such spikes in frequency previously, which anecdotally have coincided with stressful events such as shipment of animals or rearrangement of paired animals in a room, likely leading to CMV reactivation events *in vivo* followed by expansion of CMV-specific T cells. However, we do not have any evidence to explain the specific spike at day 56 in CD4+ T cells.

REVIEWERS' COMMENTS

Reviewer #1 (Remarks to the Author):

The authors revised the manuscript and expanded the discussion. I have several questions.

1. In Fig. 1C, how were T cells in the BAL recruited after subcutaneous immunization?
2. Does an M gene contain M1 and M2 genes? If the M2 gene is included, antibody against M2 should be examined. How did antibody against M2 contribute to protection in this study?
3. In addition, a mechanism that how CD4+ TEM cells suppress viral titers and improved the survival rate should be discussed.

Reviewer #2 (Remarks to the Author):

The authors have addressed all comments raised.

Reviewer #3 (Remarks to the Author):

The authors have addressed the comments raised during review

Response to Reviewers:

Author responses in blue.

Reviewer #1 (Remarks to the Author):

The authors revised the manuscript and expanded the discussion. I have several questions.

1. In Fig. 1C, how were T cells in the BAL recruited after subcutaneous immunization?

CMV-specific T cells are known to accumulate at mucosal sites, including the lung. Indeed, as stated on lines 88-91, this phenomenon partially formed the rationale for this vaccine approach. There exist two likely mechanisms by which CMV-induced T cells accumulate in high frequencies in the lung. First, effector memory T cells traffic to and persist in extralymphatic sites such as the lung. Therefore, following priming by CMV these effector memory T cells traffic to sites of inflammation such as the lung. Secondly, although the CMV vaccine vectors used here are spread deficient, they do still spread, although significantly less than the parental vector. Therefore, it is likely that this vaccine vector still does establish persistence infection in lung-resident myeloid cells, thereby priming and attracting CMV-specific T cells in the lung. We now discuss this on lines 312-314 of the manuscript.

2. Does an M gene contain M1 and M2 genes? If the M2 gene is included, antibody against M2 should be examined. How did antibody against M2 contribute to protection in this study?

We apologize for any confusion, we used only the sequence for M1, not for the overall M gene that encodes both M1 and M2. We have replaced M with M1 throughout the manuscript to clarify this point. We also state in the methods on lines 384-386 that we used M1 and thus no M2 sequences were used in the vaccines.

3. In addition, a mechanism that how CD4+ TEM cells suppress viral titers and improved the survival rate should be discussed.

We discuss mechanisms of how CD4+ TEMs protect against influenza on lines 337-344. We specifically state that we hypothesize the mechanism here is direct cytotoxic activity against influenza-infected cells. Defining this mechanism will be the focus of our next research projects for influenza.

Reviewer #2 (Remarks to the Author):

The authors have addressed all comments raised.

We thank Reviewer 3 for their overall positive assessment of our manuscript.

Reviewer #3 (Remarks to the Author):

The authors have addressed the comments raised during review

We thank Reviewer 3 for their overall positive assessment of our manuscript.